# Investigation of High-Sensitivity NO_2_ Gas Sensors with Ga_2_O_3_ Nanorod Sensing Membrane Grown by Hydrothermal Synthesis Method

**DOI:** 10.3390/nano13061064

**Published:** 2023-03-15

**Authors:** Shao-Yu Chu, Mu-Ju Wu, Tsung-Han Yeh, Ching-Ting Lee, Hsin-Ying Lee

**Affiliations:** 1Department of Photonics, National Cheng Kung University, Tainan 701, Taiwan, Republic of China; kevinvicky168@gmail.com (S.-Y.C.);; 2Department of Electrical and Electronic Engineering, Chung Cheng Institute of Technology, National Defense University, Taoyuan 335, Taiwan, Republic of China; 3Department of Electrical Engineering, Yuan Ze University, Taoyuan 320, Taiwan, Republic of China

**Keywords:** field emission scanning electron microscope, Ga_2_O_3_ nanorods, hydrothermal synthesis method, NO_2_ gas sensors, X-ray diffraction, X-ray photoelectron spectroscopy

## Abstract

In this work, Ga_2_O_3_ nanorods were converted from GaOOH nanorods grown using the hydrothermal synthesis method as the sensing membranes of NO_2_ gas sensors. Since a sensing membrane with a high surface-to-volume ratio is a very important issue for gas sensors, the thickness of the seed layer and the concentrations of the hydrothermal precursor gallium nitrate nonahydrate (Ga(NO_3_)_3_·9H_2_O) and hexamethylenetetramine (HMT) were optimized to achieve a high surface-to-volume ratio in the GaOOH nanorods. The results showed that the largest surface-to-volume ratio of the GaOOH nanorods could be obtained using the 50-nm-thick SnO_2_ seed layer and the Ga(NO_3_)_3_·9H_2_O/HMT concentration of 12 mM/10 mM. In addition, the GaOOH nanorods were converted to Ga_2_O_3_ nanorods by thermal annealing in a pure N_2_ ambient atmosphere for 2 h at various temperatures of 300 °C, 400 °C, and 500 °C, respectively. Compared with the Ga_2_O_3_ nanorod sensing membranes annealed at 300 °C and 500 °C, the NO_2_ gas sensors using the 400 °C-annealed Ga_2_O_3_ nanorod sensing membrane exhibited optimal responsivity of 1184.6%, a response time of 63.6 s, and a recovery time of 135.7 s at a NO_2_ concentration of 10 ppm. The low NO_2_ concentration of 100 ppb could be detected by the Ga_2_O_3_ nanorod-structured NO_2_ gas sensors and the achieved responsivity was 34.2%.

## 1. Introduction

In recent years, due to the rapid development of industries in human society, environmental pollution has become increasingly serious, such as noise, air pollution, water pollution, and nuclear pollution, etc. Among them, the air pollution of nitrogen dioxide (NO_2_), causing harm to human health and the environment, is the most serious issue. Even a NO_2_ concentration of 3 ppm is enough to cause serious damage and human health problems, including throat irritation, respiratory illnesses, and even death [1,2]. Therefore, to avoid the harm caused by NO_2_ gas, it is very important to develop a gas sensor with high responsivity and high selectivity to detect NO_2_ gas.

Among several structures of gas sensors, a metal oxide semiconductor (MOS) gas sensor is the most attractive structure due to its inherent advantages of easy fabrication, simple operation, low prices, and a small size [3]. Many metal oxide semiconductor materials have played promising roles in resistive types of MOS-structured gas sensors, such as zinc oxide (ZnO) [4,5], stannic oxide (SnO_2_) [6,7], titanium dioxide (TiO_2_) [8,9], indium oxide (In_2_O_3_) [10,11], and gallium oxide (Ga_2_O_3_) [12,13]. Among the metal oxide semiconductor materials, in view of the advantages of non-toxicity, low prices, and good chemical stability [14,15], Ga_2_O_3_ has potential applications in high-temperature gas sensors [16,17]. Moreover, nanostructures have been designed to enhance the performance of gas sensors because of their high surface-to-volume ratio, high specific surface area, and more surface adsorption sites, recently [18,19,20,21]. In this work, the GaOOH nanorods were grown on the SnO_2_ seed layer by the hydrothermal synthesis method. The resulting GaOOH nanorods were then annealed to convert them into Ga_2_O_3_ nanorods. The surface morphology of the Ga_2_O_3_ nanorods was optimized for a high specific surface area, thereby achieving high responsivity and high selectivity in the NO_2_ gas sensors.

## 2. Materials and Methods

### 2.1. Materials

In this work, the SnO_2_ target (99.99%) with bonding on a 3 mm Cu plate was purchased from S.P. Alloys Co., Ltd., Keelung, Taiwan. Granules of gallium nitrate nonahydrate (Ga(NO_3_)_3_·9H_2_O, 99.9%) and hexamethylenetetramine (C_6_H_12_N_4_, HMT, 99.5%) were, respectively, purchased from Alfa Aesar (Heysham, UK)and Sigma-Aldrich (Darmstadt, Germany). A target-gas NO_2_ gas cylinder (1000 ppm) was purchased from Yun Shan Gas Co., Ltd., Tainan, Taiwan.

### 2.2. Material Characterization

X-ray diffraction (XRD, D8 DISCOVER with GADDS, Bruker AXS Gmbh, Karlsruhe, Germany) was used to characterize the seed layers of Ga_2_O_3_ nanorods. The morphological and structural analyses of the resulting Ga_2_O_3_ nanorods were performed with a field emission scanning electron microscope (FE-SEM, AURIGA, ZEISS, Oberkochen, Germany). The material characteristics of the annealing-treated GaOOH nanorods were measured using X-ray photoelectron spectroscopy (XPS, PHI 5000 VersaProbe III, ULVAC-PHI. Inc., Osaka, Japan). The current–voltage (I-V) characteristics were obtained with an Agilent 4156C (Santa Clara, CA, USA) semiconductor parameter analyzer.

### 2.3. Experimental Details

Figure 1 shows the schematic configuration of the Ga_2_O_3_ nanorod-structured NO_2_ gas sensors. A radio frequency (RF) magnetron sputtering system was used to deposit SnO_2_ films with various thicknesses of 50 nm, 100 nm, and 200 nm on quartz substrates as seed layers for GaOOH nanorods. The RF power, the Ar/O_2_ gas ratio, and the chamber pressure were maintained at 75 W, 48/2 sccm, and 5 mtorr, respectively. The growth rate of the SnO_2_ films was approximately 6.8 nm/min. After completing the various SnO_2_ seed layers, the GaOOH nanorods were grown on quartz substrates using the hydrothermal synthesis method with various concentration solutions consisting of Ga(NO_3_)_3_·9H_2_O and HMT at 180 °C for 4 h using a magnetic stirrer hotplate. The nanorods were then converted from GaOOH to Ga_2_O_3_ in a pure N_2_ ambient atmosphere using a furnace system for 2 h at 300 °C, 400 °C, and 500 °C, respectively. The hydrothermal synthesis processes and the material conversion during the thermal annealing processes are described in Equations (1–4), respectively [22].
C_6_H_12_N_4_ + 6H_2_O → 6HCNO + 4NH_3_(1)
NH_3_ + H_2_O → NH^4+^ + OH^−^(2)
Ga^3+^ + 3OH^−^ → GaOOH + H_2_O(3)
2GaOOH → Ga_2_O_3_ + H_2_O(4)

The Ni/Au (20/100 nm) metals were deposited on the Ga_2_O_3_ nanorods as the electrodes of the gas sensors by an electron-beam evaporator.

Figure 2 shows the schematic configuration of the measurement system of the NO_2_ gas sensors. A target-gas NO_2_ gas cylinder and a mass flow controller (MFC) were installed with a closed chamber to provide a stable NO_2_ gas source, and an Agilent 4156C semiconductor parameter analyzer was equipped to measure the current–voltage (I-V) characteristics of the Ga_2_O_3_ nanorod-structured NO_2_ gas sensors. In addition, the closed chamber of the NO_2_ gas sensor measurements was equipped with a humidity controller to maintain a relative humidity of 30% during the testing process. When NO_2_ gas was introduced into the chamber, the NO_2_ molecules would react with the Ga_2_O_3_ sensing membrane, causing an increase in the resistance of the sensor. This reaction occurred as a result of the NO_2_ molecules extracting electrons from the Ga_2_O_3_ nanorod sensing membrane, which reacted with the O_2_^−^_(abs)_ in the sensing membrane. These reaction processes were as follows in Equations (5)–(8) [23].
O_2(gas)_ → O_2(abs)_(5)
O_2(abs)_ + e^−^ → O_2_^−^_(abs)_(6)
NO_2(gas)_ + e^−^ → NO_2_^−^_(abs)_(7)
NO_2(gas)_ + O_2_^−^_(abs)_ + 2e^−^→ NO_2_^−^_(abs)_ + 2O_2_^−^_(abs)_(8)

When the NO_2_ gas was removed from the chamber and purged by air, the electrons previously trapped by NO_2_ molecules were released back to the conductive band of the material, leading to a decrease in sensor resistance and a return to its initial state.

## 3. Results

Since the diameter of the resulting nanorods was dependent on the average grain size of the seed layer, which was increased with an increase in the film thickness [24,25,26], an amorphous seed layer with a small grain size has become a very important research target to obtain nanorods with a larger surface-to-volume ratio. In this work, to achieve a high surface-to-volume ratio, high specific surface area, and more surface adsorption sites, the surface morphology was optimized by changing the thickness of the SnO_2_ seed layer and the mixed solution concentration of the hydrothermal precursors.

Using the measurement of X-ray diffraction (XRD) with CuKα radiation, Figure 3 illustrates the crystalline characteristics of the SnO_2_ seed layers with various thicknesses of 50 nm, 100 nm, and 200 nm. As shown in Figure 3, the 50-nm-thick SnO_2_ film did not have any obvious peak in the XRD pattern. Furthermore, for the 100-nm-thick and 200-nm-thick SnO_2_ films, three diffraction peaks located at 26.5°, 34.0°, and 51.9°, corresponding to the SnO_2_ (110), (101), and (211) planes, respectively, were found [27]. It could be observed that the crystallinity of the SnO_2_ film was improved with an increase in the SnO_2_ thickness. Consequently, to enable the growth of GaOOH nanorods with a high surface-to-volume ratio morphology, the amorphous structure of a 50-nm-thick SnO_2_ film was required for the seed layer in this study.

Figure 4a–c illustrate the FE-SEM top-view and cross-section images of the GaOOH nanorods respectively synthesized on the various SnO_2_ seed layers by the hydrothermal synthesis method with Ga(NO_3_)_3_·9H_2_O and HMT concentrations of 12 mM and 10 mM. It was seen that the shape of the nanorods was approximately a rhombus and the dimension of the nanorods showed a uniform distribution. However, the size and number of the nanorods were obviously influenced by the thickness of the SnO_2_ seed layers. For the GaOOH nanorods grown on the SnO_2_ seed layers with various thicknesses of 50 nm, 100 nm, and 200 nm, the average short-side diagonal was approximately 52.0 nm, 60.2 nm, and 72.6 nm, respectively. The average short-side diagonal of the GaOOH nanorods was increased with an increase in the thickness of the SnO_2_ seed layers. The height of the resulting nanorods was almost kept at around 320 nm, with no significant difference. Consequently, it could be deduced that the morphology of the GaOOH nanorods was greatly influenced by the thickness of the seed layer. A larger surface-to-volume ratio of GaOOH nanorods was obtained in the thinner SnO_2_ seed layer due to the smaller grain size of the thinner seed layer.

In the hydrothermal synthesis processes, to obtain an optimized morphology of GaOOH nanorods by changing the concentration of Ga(NO_3_)_3_·9H_2_O and HMT precursors, the various Ga(NO_3_)_3_·9H_2_O/HMT precursor concentrations of 6 mM/5 mM, 12 mM/10 mM, and 18 mM/15 mM were utilized and investigated. The FE-SEM top-view and cross-section images of the various GaOOH nanorods grown on the 50-nm-thick SnO_2_ seed layer are shown in Figure 5a–c. The average short-side diagonal of the GaOOH nanorods grown using various Ga(NO_3_)_3_·9H_2_O/HMT precursor concentrations of 6 mM/5 mM, 12 mM/10 mM, and 18 mM/15 mM was 51.5 nm, 52.0 nm, and 70.7 nm, respectively. The corresponding height of the GaOOH nanorods was approximately 195 nm, 320 nm, and 352 nm, respectively. It was found that the average short-side diagonal of the GaOOH nanorods gradually increased with an increase in the concentration of the Ga(NO_3_)_3_·9H_2_O/HMT precursor. Although the smallest average short-side diagonal was obtained in the GaOOH nanorods grown using the precursor concentration of 6 mM/5 mM, the associated nanorods density and height were significantly lower than those grown with the other precursor concentrations. This phenomenon was attributed to the fact that the reactants using 6 mM/5 mM precursors were not sufficient in concentration to deposit at every site where the GaOOH nanorods could be formed. Consequently, the most suitable synthesis conditions for the GaOOH nanorods were the 50-nm-thickness SnO_2_ seed layer and the Ga(NO_3_)_3_·9H_2_O/HMT precursor concentration of 12 mM/10 mM, which exhibited the largest surface-to-volume ratio for the GaOOH nanorods.

To improve the gas sensitivity of the NO_2_ gas sensors, the GaOOH nanorods should be converted into Ga_2_O_3_ nanorods by annealing treatment. In addition, XPS was carried out to study the existence of oxygen vacancies (O_vacancy_) and -OH bonds in the Ga_2_O_3_ nanorod sensing membranes with various annealing temperatures. Figure 6a–d show the O1s core level spectra of the GaOOH nanorods without and with annealing treatment for 2 h at 300 °C, 400 °C, and 500 °C, respectively. The O1s peak was composed of three bands located at the binding energy of 530.8 eV, 532.1 eV, and 533.0 eV, which were, respectively, assigned to the Ga^3+^, O_vacancy_, and -OH bonds [28]. According to the XPS results, the peak intensity of the -OH bonds was significantly reduced when increasing the annealing temperature. This phenomenon indicated that the thermal energy in the annealing treatment process could cause the dehydroxylation reaction in the Ga_2_O_3_ nanorods [29]. Moreover, the peak ratio of the Ga^3+^ and O_vacancy_ bonds (Ga^3+^/O_vacancy_) for the GaOOH nanorods without and with annealing treatment for 2 h at 300 °C, 400 °C, and 500 °C was 11.93, 8.43, 7.38, and 7.01, respectively. It is worth noting that the oxygen vacancies could be effectively increased on the surfaces of the Ga_2_O_3_ nanorods in the annealing treatment process, thereby increasing the active sites for the NO_2_ gas.

Figure 7 shows the temperature dependence of the resistance (R_S_(T)) for the NO_2_ gas sensors with Ga_2_O_3_ nanorod sensing membranes annealed at various temperatures. In general, the reaction rate and operating temperature were mainly affected by the activation energy (E_A_) in the gas sensors. The activation energy of the Ga_2_O_3_ nanorod-structured NO_2_ gas sensors was calculated using the following Arrhenius equation [30]:(9)$$R_{s}\left(T\right)=R_{0}e^{\frac{E_{A}}{\mathrm{kT}}}$$
(10)$$\ln\left[R_{s}\left(T\right)\right]=\ln\left(R_{0}\right)+\left(\frac{E_{A}}{1000k}\right)\left(\frac{1000}{T}\right)$$
where R_0_ is the pre-exponential factor, k is the Boltzmann constant, and T is the absolute temperature, respectively. The activation energy is determined by the slope of the Arrhenius plot. As shown in Figure 7, the activation energy of 248 meV, 214 meV, and 211 meV corresponded to the NO_2_ gas sensors with Ga_2_O_3_ nanorod sensing membranes annealed at 300 °C, 400 °C, and 500 °C, respectively. In general, the activation energy was inversely proportional to the carrier concentration [31,32], which was also increased with the amounts of oxygen vacancies in the Ga_2_O_3_ material [33]. According to the XPS results, the oxygen vacancies residing on the surfaces of Ga_2_O_3_ nanorods were effectively increased during the annealing treatment process. Consequently, it could be deduced that the activation energy was decreased with an increase in the annealing temperature.

In this study, the gas responsivity (R) was calculated using the following equation:(11)$$R\left(\%\right)=\frac{R_{g}-R_{a}}{R_{a}}\times 100\%$$
where R_g_ and R_a_ are the resistances of the NO_2_ gas sensors in NO_2_ gas and air environments, respectively. Figure 8 shows the responsivity versus operating temperature characteristics of the NO_2_ gas sensors with Ga_2_O_3_ nanorod sensing membranes annealed at various temperatures. Under a bias voltage of 1 V and a NO_2_ gas concentration of 10 ppm, the optimal responsivity of the NO_2_ gas sensors with Ga_2_O_3_ nanorod sensing membranes annealed at 300 °C, 400 °C, and 500 °C was 225.8%, 1184.6%, and 824.9%, respectively. The corresponding operating temperatures of the optimal responsivity were 300 °C, 275 °C, and 250 °C. It could be found that the responsivity increased with an increase in the annealing temperature until 400 °C and then decreased when further increasing the annealing temperature to 500 °C. The reduction in operating temperature tendency was followed by the activation energy tendency of the annealing temperature of the Ga_2_O_3_ nanorod sensing membranes. The enhanced responsivity was attributed to the fact that the annealing thermal energy could effectively increase the oxygen vacancies, thereby providing more gas-reactive surface sites. However, the improvement responsivity of the NO_2_ gas sensors with 500 °C-annealed Ga_2_O_3_ nanorod sensing membranes was degraded. It could be deduced that the induced excessive carrier concentration present in the 500 °C-annealed Ga_2_O_3_ nanorod sensing membranes could reduce the resistance variation [34].

Figure 9a,b show the response time (τ_r_) and recovery time (τ_d_) of the NO_2_ gas sensors with Ga_2_O_3_ nanorod sensing membranes annealed at various temperatures under a bias voltage of 1 V and a NO_2_ gas concentration of 10 ppm at their associated individual optimal operating temperatures. In general, the response time and recovery time were calculated as the time from 0% to 90% of the maximum responsivity and from 100% to 10% of the maximum responsivity, respectively [35]. As shown in Figure 9a,b, under the individual optimal operating temperatures of 300 °C, 275 °C, and 250 °C, the response time of the NO_2_ gas sensors with Ga_2_O_3_ nanorod sensing membranes annealed at 300 °C, 400 °C, and 500 °C was 75.7, 63.6, and 61.5 s, respectively. The corresponding recovery time was 410.4, 135.7, and 125.9 s, respectively. These results indicated that both the response time and the recovery time of the resulting NO_2_ gas sensors decreased with an increase in the annealing temperature. This phenomenon was attributed to the fact that lower activation energy of the Ga_2_O_3_ nanorod-structured gas sensors could be achieved by annealing the sensing membranes at a higher temperature. The reduction in the response time and the recovery time of the resulting gas sensors could be induced by the lower activation energy [36,37].

Figure 10 presents the dynamic gas responsivity of the NO_2_ gas sensors with 400 °C annealed-Ga_2_O_3_ nanorod sensing membranes under various NO_2_ concentrations at an operating temperature of 275 °C. It could be found that the responsivity of the gas sensor increased with an increase in the NO_2_ concentration, reaching saturation at 50 ppm. The NO_2_ gas sensors with 400 °C annealed-Ga_2_O_3_ nanorod sensing membranes could be effectively detected even at a very low NO_2_ concentration of 100 ppb and the achieved responsivity was 34.2%.

To investigate the gas selectivity of the NO_2_ gas sensor with a Ga_2_O_3_ nanorod sensing membrane, alcohol (C_2_H_5_OH) and ammonia (NH_3_) were also used as the target gases in this study. Figure 11 shows the responsivity of the gas sensor with a 400 °C-annealed Ga_2_O_3_ nanorod sensing membrane under various target gases. Under a C_2_H_5_OH concentration of 100 ppm, a NH_3_ concentration of 100 ppm, and a NO_2_ concentration of 10 ppm, the responsivity of the NO_2_ gas sensors was 154.8%, 187.2%, and 1184.6%, respectively, at an operating temperature of 275 °C. This indicated that the gas sensor had certain sensing capability for C_2_H_5_OH and NH_3_ gases, but the responsivity was significantly lower than that of NO_2_ gas. Consequently, it could be concluded that the NO_2_ gas sensors using the 400 °C-annealed Ga_2_O_3_ nanorods as the sensing membranes had high selectivity for NO_2_ gas.

To further highlight the results of this work, the performance of the NO_2_ gas sensors using Ga_2_O_3_ nanorods as the sensing membranes were compared with other similar studied NO_2_ gas sensors reported previously, as listed in Table 1. The performance of the NO_2_ gas sensors with Ga_2_O_3_ nanorod sensing membranes exhibited excellent features.

## 4. Conclusions

In this study, various Ga_2_O_3_ nanorods were successfully grown on quartz substrates as sensing membranes of NO_2_ gas sensors using the hydrothermal synthesis method and annealing processes. To increase the surface-to-volume ratio of the GaOOH nanorods, the thickness of the SnO_2_ seed layer and the concentration of the hydrothermal precursor (Ga(NO_3_)_3_·9H_2_O/HMT) were optimized. The surface-to-volume ratio of the GaOOH nanorods decreased with an increase in the thickness of the SnO_2_ seed layer due to the reduction in the resulting grain size. Moreover, by decreasing the concentration of the hydrothermal precursor (Ga(NO_3_)_3_·9H_2_O/HMT), the surface-to-volume ratio of the resulting GaOOH nanorods gradually increased. Although the concentration of 6 mM/5 mM had the highest surface-to-volume ratio, its nanorod density was significantly lower than for the other concentrations. It was found that the surface-to-volume ratio of the GaOOH nanorods could be effectively optimized using the 50-nm-thick SnO_2_ seed layer and the Ga(NO_3_)_3_·9H_2_O/HMT concentration of 12 mM/10 mM. The dependence of oxygen vacancies on the annealing temperature of Ga_2_O_3_ nanorods was verified by the measurements of the XPS experimental results. When the Ga_2_O_3_ nanorods were annealed at various temperatures, the amounts of oxygen vacancies were increased and the number of -OH bonds was suppressed by the thermal treatment. Consequently, the associated activation energy of the NO_2_ gas sensors was decreased from 248 meV to 211 meV when increasing the annealing temperature of the Ga_2_O_3_ nanorod sensing membranes from 300 °C to 500 °C. The Ga_2_O_3_ nanorod sensing membrane annealed for 2 h at 400 °C achieved the maximum responsivity of 1184.6%. The response time and recovery time of the NO_2_ gas sensors with Ga_2_O_3_ nanorod sensing membranes were effectively improved by the annealing treatment, which was due to the activation energy tendency of the Ga_2_O_3_ nanorods. Furthermore, because the Ga_2_O_3_ nanorod-structured NO_2_ gas sensor revealed high sensitivity, it could even detect NO_2_ gas with a concentration as low as 100 ppb. Moreover, the gas sensor also exhibited high selectivity towards NO_2_ gas, and the responsivity of the gas sensors under the NO_2_ concentration of 10 ppm was larger than that under the C_2_H_5_OH and NH_3_ concentrations of 100 ppm. Consequently, it is verified that the low-cost hydrothermal synthesis method can grow GaOOH nanorods that can be converted into Ga_2_O_3_ nanorods using a thermal annealing process. The resulting Ga_2_O_3_ nanorods are promising candidates for NO_2_ gas sensors.

## Figures and Tables

**Figure 1 nanomaterials-13-01064-f001:**
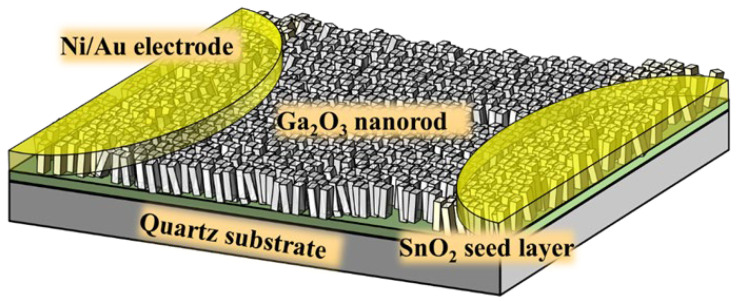
Schematic configuration of NO_2_ gas sensors with Ga_2_O_3_ nanorod sensing membrane.

**Figure 2 nanomaterials-13-01064-f002:**
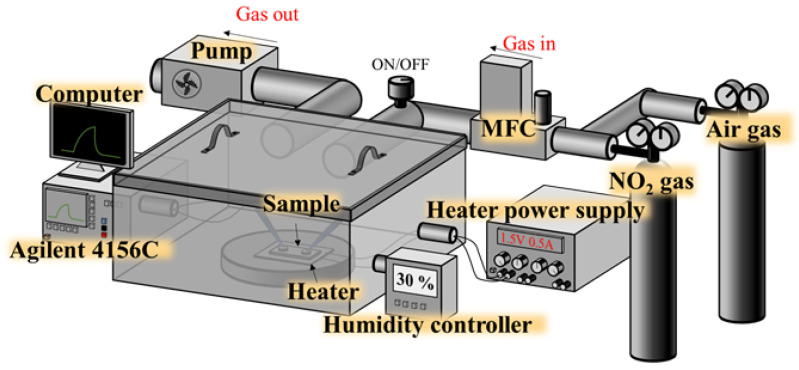
Schematic configuration of measurement system of NO_2_ gas sensors.

**Figure 3 nanomaterials-13-01064-f003:**
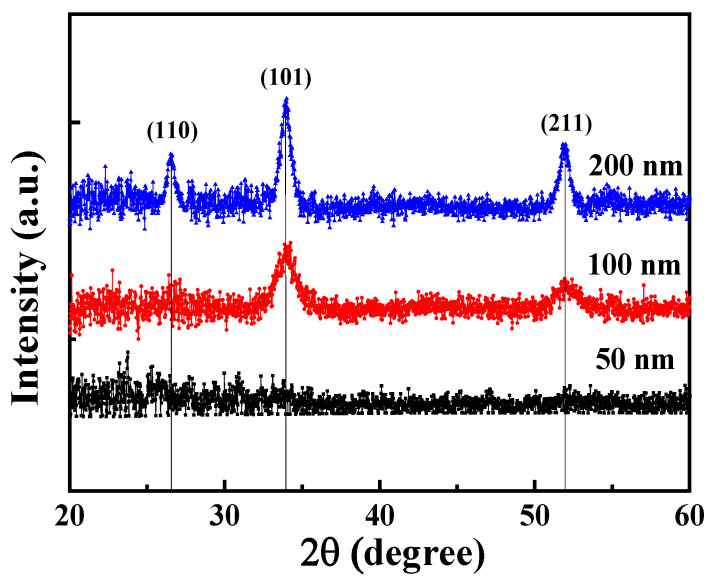
XRD spectra of SnO_2_ seed layers with various thicknesses.

**Figure 4 nanomaterials-13-01064-f004:**
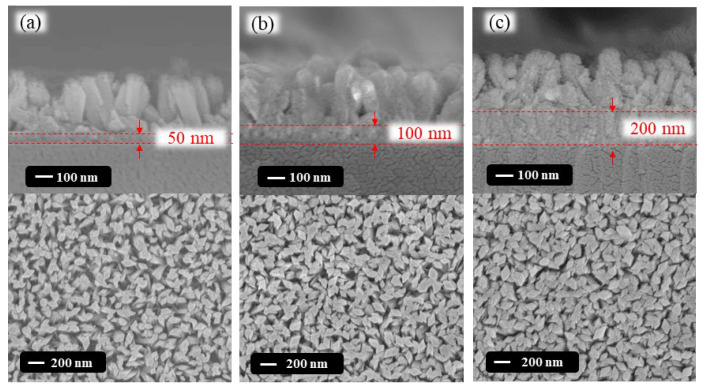
FE-SEM top-view and cross-section images of GaOOH nanorods grown on SnO_2_ seed layers with a thickness of (**a**) 50 nm, (**b**) 100 nm, and (**c**) 200 nm.

**Figure 5 nanomaterials-13-01064-f005:**
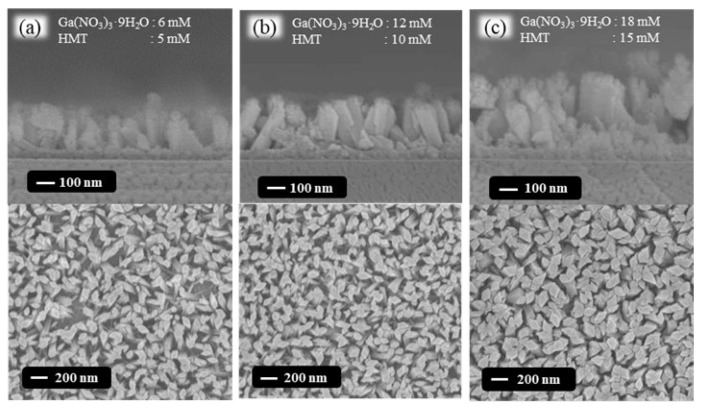
FE-SEM top-view and cross-section images of GaOOH nanorods grown using various precursor concentrations of (**a**) 6 mM/5 mM, (**b**) 12 mM/10 mM, and (**c**) 18 mM/15 mM.

**Figure 6 nanomaterials-13-01064-f006:**
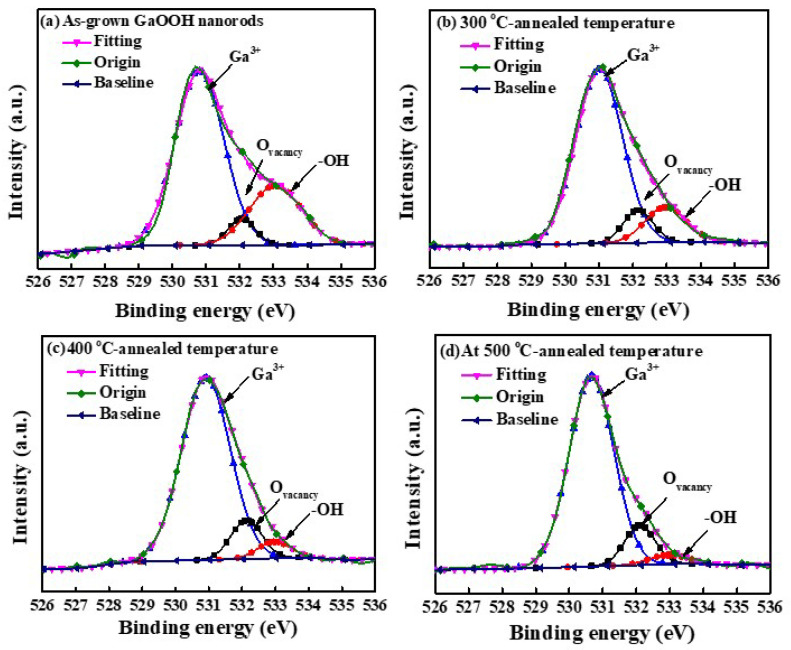
XPS spectra of O1s core-level spectra of (**a**) as-grown GaOOH nanorods and annealed Ga_2_O_3_ nanorods treated at (**b**) 300 °C (**c**) 400 °C, and (**d**) 500 °C.

**Figure 7 nanomaterials-13-01064-f007:**
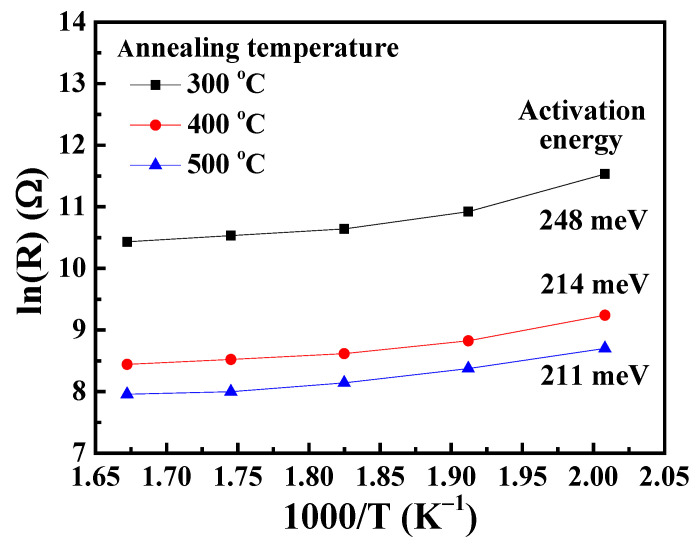
Temperature dependence of resistance for NO_2_ gas sensors with Ga_2_O_3_ nanorod sensing membranes annealed at various temperatures.

**Figure 8 nanomaterials-13-01064-f008:**
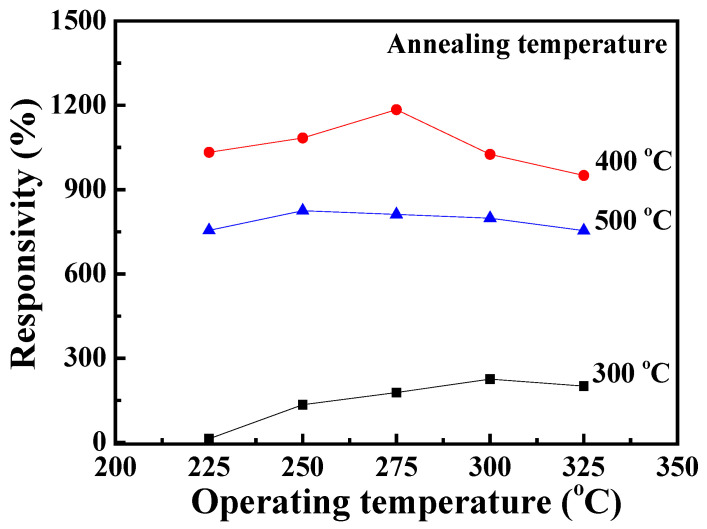
Responsivity versus operating temperature of NO_2_ gas sensors with Ga_2_O_3_ nanorod sensing membranes annealed at various temperatures.

**Figure 9 nanomaterials-13-01064-f009:**
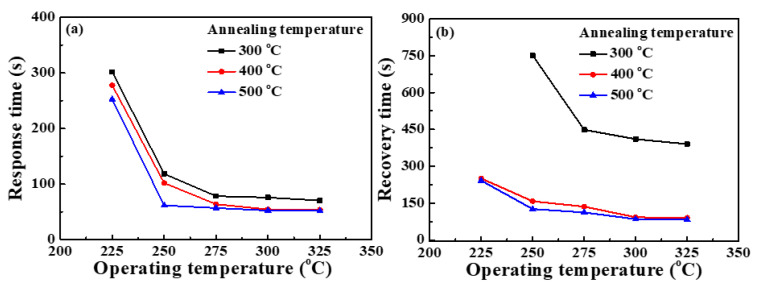
(**a**) Response time and (**b**) recovery time of NO_2_ gas sensors with Ga_2_O_3_ nanorod sensing membranes annealed at various temperatures under 10 ppm NO_2_ gas concentration.

**Figure 10 nanomaterials-13-01064-f010:**
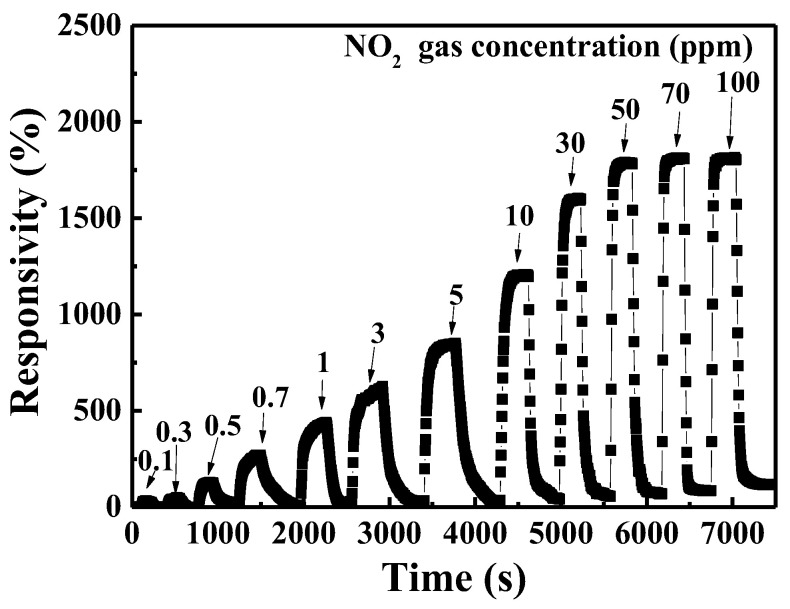
Dynamic gas responsivity of NO_2_ gas sensor with 400 °C-annealed Ga_2_O_3_ nanorod sensing membrane under various NO_2_ concentrations at an operating temperature of 275 °C.

**Figure 11 nanomaterials-13-01064-f011:**
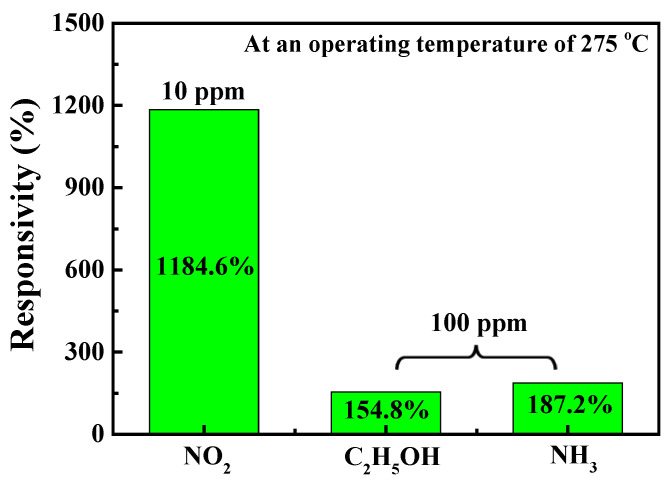
Responsivity of gas sensor with 400 °C-annealed Ga_2_O_3_ nanorod sensing membrane under various target gases.

**Table 1 nanomaterials-13-01064-t001:** Performance comparison of various structured NO_2_ gas sensors.

Materials and Structure	Responsivity	Operating Temperature (°C)	Minimum Concentration of NO_2_ (ppm)	Ref.
TiO_2_-Ga_2_O_3_ thin film	~2.4	200	0.5	[38]
ZGO thin film	1.18	300	0.5	[39]
Oxidized galinstan	~1.8%	100	1	[40]
ZnO/ZnS core–shell nanowires	293.29%	300	1	[41]
ZnO nanowalls	9.63	220	5	[42]
Cu-doped ZnO thin film	26%	200	5	[43]
Ga_2_O_3_ nanorods	34.2%	275	0.1	This work

## Data Availability

The data presented in this study are available on request from the corresponding author.

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
