# Peer review of "Investigation of High-Sensitivity NO2 Gas Sensors with Ga2O3 Nanorod Sensing Membrane Grown by Hydrothermal Synthesis Method"

_nanomaterials, 2023, doi:10.3390/nano13061064_

Round 1

Reviewer 1 Report

The manuscript ID: 2278357 " Investigation of high-sensitivity NO2 gas sensors with Ga2O3 nanorod sensing membrane grown by hydrothermal synthesis method" presents a NO2 gas sensor based on Ga2O3 nanorod, where is need a high temperature to get a response.  In my opinion, This study needs a mayor revision

1. The title of the manuscript should be modified because the authors don´t demonstrate that the sensor responds to selectivity to NO2. A selectivity study should be included in the manuscript to demonstrate that the sensor responds to NO2 selectively, otherwise a sensor for NO2 cannot be considered. A selectivity study should be included in the manuscript to demonstrate that the sensor responds to NO2 selectively, otherwise a sensor for NO2 cannot be considered.

2. In Abstract. the response of the sensor is established with %, as it is possible to obtain responses of 1184.6%!!!!. This is an error.

3. In the last sentence of the abstract "The low NO2 concentration of 100 ppb could be detected by.....". What are the authors basing themselves on to make that statement? If the authors have an explanation that demonstrates that phrase, it is necessary that they indicate it, otherwise, that phrase has to be eliminated.

4. The introduction must be modified, especially the last part, since there are sensors based on gadolinium oxide that respond to room temperature in the bibliography, the authors must emphasize that what is new is what they achieve with the proposed sensor since it is not clear In the introduction.

5. Pg 6, Sensor responses of 225.8%, 1184.6%, and 824.9% are indicated, those values cannot exceed 100%.

6. Figure 9 is not correct. Response and recovery times are not seen on that graph. A new graph indicating response and recovery times should be included.

7. Figure 10. What is the temperature with which you are working 400ºC or 275 ºC?, it is not clear in the text on page 7.

8. What does it mean that saturation is reached at 50 ppm? In figure 10 the authors include a response up to 100 ppm. In case there are no differences between the response at 50 ppm and at 100 ppm, the authors should include an explanation of what would be happening.

9.  Table 1 presents serious errors. Authors should check the Response (%) column, and should also highlight the sensors that respond to room temperature. In addition, a discussion about the results found should be included in the text.

10. A selectivity study must be carried out and included, regarding the main gases that are in the atmosphere and possible pollutants.

11. When the authors resolve all the doubts raised, they will probably have to reconsider rewriting the conclusions.  

Author Response

請參閱附件。

Reviewer 2 Report

There are some elements that the authors should clarify in the manuscript.

- XRD spectra of SnO2 seed layer on Fig 2 is not edifying. Some more relevant material spectral markers would be indicated to be mentioned on the figure.

- The operating temperature of the sensors is very close to the treatment (annealing) temperature, which is why they will not have reproductive responses because any measurement will bring an extra treatment etc. In this respect, some reproducibility measurements or at least an analysis of these aspects would be useful to be pointed out because over time the sensors will change their parameters (see https://doi.org/10.1016/j.matchemphys.2022.126691)

- The effect of humidity on the nanorod surface and implicitly the humidity dependence of the sensors' response must be mentioned, or at least the humidity conditions in which the measurements were made should be mentioned.

- Regarding the quality of the presentation, the figures must have a unitary aspect, the text must (re)written more carefully etc.

Reviewer 3 Report

The article is devoted to the development of a NO2 sensor based on Ga2O3 nanorods. The high sensitivity of the sensors is demonstrated by optimizing the size of the nanorods. A decrease in diameter leads to an increase in the working surface of the sensor. The effect of SnO2 seed film thickness size and precursor concentration on the diameter of Ga2O3 nanorods was determined. The article is well written, but a number of unresolved issues remain:

1. In Table 1, it is difficult to give preference to ZnO or Ga2O3 nanorods. It would be interesting to make a comparison on the detection of the minimum concentration of NO2.

2. It is necessary to present the mechanism of NO2 absorption by Ga2O3 nanorods.

3. It is reasonable to determine the saturation limit of Ga2O3 nanorods during NO2 absorption. And how to restore the functioning of the sensor afterwards.

4. The article presents an example of a resistive sensor. Can be compared with the resonance sensor, when the nanorods form on the surface of QCM. Then an increase in the mass of the nanorods leads to a decrease in the resonance frequency.

05.03.2023

Round 2

Reviewer 1 Report

The authors have clarified some doubts for me. As a recommendation, the authors should inspect Table 1 has errors, please check the values.

It would be interesting if the authors included a new Figure 10 Resistance vs time. 

Reviewer 2 Report

Ok

Author Response

We greatly appreciate your suggestions.

Reviewer 3 Report

The authors made the recommended changes to the article. The article can be published. The article has a good scientific level.

Author Response

We greatly appreciate your suggestions.